# Utilization of Industrial By-Products/Waste to Manufacture Geopolymer Cement/Concrete

**Numanuddin M. Azad** [1] **and S.M. Samindi M.K. Samarakoon** [2,*]

1 Structural Engineering, Rashtrasant Tukadoji Maharaj Nagpur University, Nagpur 440033, India; naumanuddin.azad@gmail.com
2 Department of Mechanical and Structural Engineering and Material Science, University of Stavanger, 4021 Stavanger, Norway
* Correspondence: samindi.samarakoon@uis.no; Tel.: +47-518-32387

**Abstract:** There has been a significant movement in the past decades to develop alternative sustainable building material such as geopolymer cement/concrete to control $CO_2$ emission. Industrial waste contains pozzolanic minerals that fulfil requirements to develop the sustainable material such as alumino-silicate based geopolymer. For example, industrial waste such as red mud, fly ash, GBFS/GGBS (granulated blast furnace slag/ground granulated blast furnace slag), rice husk ash (RHA), and bagasse ash consist of minerals that contribute to the manufacturing of geopolymer cement/concrete. A literature review was carried out to study the different industrial waste/by-products and their chemical composition, which is vital for producing geopolymer cement, and to discuss the mechanical properties of geopolymer cement/concrete manufactured using different industrial waste/by-products. The durability, financial benefits and sustainability aspects of geopolymer cement/concrete have been highlighted. As per the experimental results from the literature, the cited industrial waste has been successfully utilized for the synthesis of dry or wet geopolymers. The review revealed that that the use of fly ash, GBFS/GGBS and RHA in geopolymer concrete resulted high compressive strength (i.e., 50 MPa–70 MPa). For high strength (>70 MPa) achievement, most of the slag and ash-based geopolymer cement/concrete in synergy with nano processed waste have shown good mechanical properties and environmental resistant. The alkali-activated geopolymer slag, red mud and fly ash based geopolymer binders give a better durability performance compared with other industrial waste. Based on the sustainability indicators, most of the geopolymers developed using the industrial waste have a positive impact on the environment, society and economy.

**Keywords:** sustainability; industrial wastes; geopolymer concrete; pozzolanic binders; building materials; alkali-activation; sustainability indicator

## 1. Introduction

It is a well-known fact that, in every construction activity, cement plays a key role as a pozzolanic binder. The reaching of an alerting stage in global warming, due to the emission of greenhouse gases (GHG), has shifted the attention of researchers towards green energy-based construction practices, since the production of cement is a highly energy-intensive process. Hence, in the search for alternative construction materials, researchers found the geopolymer process more feasible, as per its technological and environmental benefits [1,2] because manufacturing 1 ton of cement emits approximately 0.7 tons of $CO_2$ into the atmosphere [3]. The concept of geopolymers was first adopted by Prof. Joseph Davidovits in 1978. These are the alumino-silicates that are formed with the dissolution of amorphous silica and alumina in a highly alkali-activated chemical activator medium at ambient temperature. According to Prof. Sir Joseph Davidovits, strong alkalis such as NaOH (sodium hydroxide) and KOH (potassium hydroxide) are efficient to carry dissolving reaction for silica ($SiO_2$) and alumina ($Al_2O_3$) to form alumino-silicates [4]. Additionally, this reaction is

mainly influenced by the raw material characteristics, activator concentration, and curing process (drying time and temperature). Alkalis are readily available in the market, and the source of silica and alumina in abundance had been spotted by researchers in various industrial wastes [5]. Many industrial sectors, like aluminium, steel, power plants and biomass, have conserved these mineral values in the form of wastes. In aluminium, industrial waste such as red mud (RM), PLK (partially laterite kondalite) and KK (kaolinitic kondalite) are rich in minerals that can fulfil the requirement of the geopolymer process. Steel industrial waste has an abundance of calcium and silica minerals, which are the key minerals required for making different silicates of calcium and their mineralogical calcium silicate phases [6,7]. Biomass waste such as rice husk and bagasse can be converted into ashes, which show approximately 80–90% of silica in amorphous form. This amorphous form of silica ($SiO_2$) can be a cost-cutter for geopolymer processes [8]. It is also well known that power plants' burnt waste ash, like fly ash and bottom ash, also contains good pozzolanic properties, which are being utilized in cement production as a 30% partial replacement. However, through the process of geopolymerization, 100% of fly ash can be utilized as a construction material [9]. The synergistic approach towards the use of industrial waste can set a benchmark for manufacturing construction materials which enable $CO_2$ emission to be minimized. The mechanical properties of a geopolymer binder depends on its binder ratio, type of waste material, mineralogical composition, methodology and mix design [10].

The utilization of the aforementioned industrial waste to develop geopolymer concrete/cement has a significant positive impact on the environment, society, and economy. In this case, it is vital to identify the key indicators (i.e., environmental, social and economic indicators) for sustainable building materials. In addition, the durability of new materials and manufacturing cost of geopolymer concrete should be taken into account. Moreover, the fly ash based geopolymer concrete has been manufactured with 30% more cost effective as compared to the conventional cement concrete. Whereas, the compressive strength has reached up to 62 MPa while OPC based concrete showed up to 21 MPa [11]. In addition, energy and cost analysis of geopolymer brick synthesis in comparison with conventional bricks was found more economical as the production gained 5% profit for the development of slag-based geopolymer bricks [12]. In case of embodied energy of the geopolymer, the fly ash based manufactured geopolymer synthesis showed 40% less energy requirement as compared to cement-based concrete. However, the chemical activator ingredients for alkali activation consumes 39% energy for sodium hydroxide and 49% energy for sodium silicate, which shows that the alternatives for alkali have to be detected from some other liquid waste should have to be searched to make it more economical [13]. However, the low-cost solution for geopolymer synthesis has been accelerated to find replacement to alkali-activators through industrial liquid waste fulfilling the alkali activation demand. The Bayer's liquid from extraction of alumina could be a cost-effective solution, as this liquid waste contains sufficient number of alumino-silicates in the form of geopolymer precursor [14].

The geopolymer research scenario could fulfil the sustainability requirements to meet the construction materials' requirement. However, the industrial waste from cited industrial origins have the potential to cover the construction material requirement by adopting the geopolymer process. The raw material availability, process output and recent synthesis findings encourage the adoption of geopolymer process for development of new sustainable building/construction materials. Therefore, it is vital to study the mechanical properties, chemical characterization, and financial benefits/sustainability of different geopolymer cements, manufactured using industrial waste/by-products, to understand the research gaps. In summary, this literature review proposes the informatory guide for civil engineers and industrial community to work together to develop these new sustainable construction materials through the geopolymer process.

## 2. Industrial Waste Materials and Their Chemical Compounds/Elements

A schematic form of a geopolymer process is shown in Figure 1, which includes the use of possible industrial wastes from metal industries like aluminium and steel, thermal plants like coal-based power plants and biomass burnt burdens like rice husk ash and bagasse ash. The process inputs are inclusive of the minerals' evaluation, on the basis of their classifications, contents and reactivity with the suitability of chemical activators.

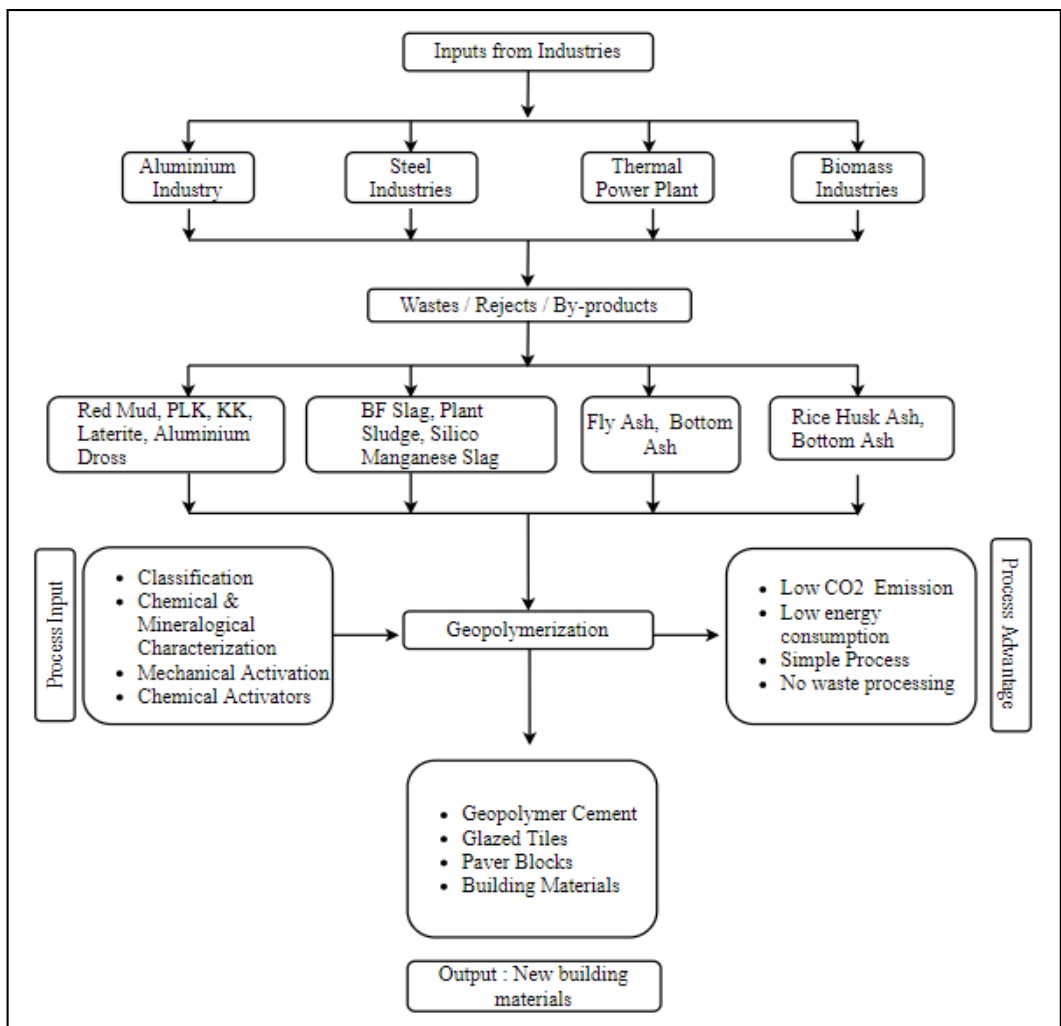

**Figure 1.** Process diagram for value-addition to industrial wastes through geopolymer synthesis.

The use of red mud-metakaolin based geopolymer is discussed in Dimas et al. [15], which states that raw material to liquid activator ratio at constant 3 M sodium hydroxide (NaOH) could be used to make red mud-metakaolin based geopolymer with the materials' composition of red mud 85% + metakaolin 15%. This finding was observed at 3.1 g/mL, which was 2.5 times more than the 2.0 g/mL ratio. Further, the change in molar concentration from 3 M to 3.5 M caused an extensive increase in the strength of materials, from almost 15 MPa to 20 MPa. An analysis of the mineral content of red mud is given in Table 1. The pozzolanic activity of the geopolymer synthesis highly depends on the major pozzolanic minerals like silica ($SiO_2$), alumina ($Al_2O_3$) and calcium oxide (CaO). It was found that the average pozzolanic content summation (PCS) of the synergy between red mud and metakaolin based geopolymer was calculated up to 87%, which was found chemically suitable for making red mud–metakaolin based geopolymer. It is observed that the ratio for the chemical compound, i.e., $SiO_2/Al_2O_3 > 2.5$, shows promising strength behaviour in geopolymer synthesis, since binding minerals like $SiO_2$ and $Al_2O_3$ chemically

react to make geopolymer alumino-silicates. The synergistic utilization of red mud with metakaolin can be effectively adopted to reduce the environmental burden of landfills.

**Table 1.** Chemical compounds in red mud, fly ash, slag, rice husk ash and bagasse ash.

| SI | Raw Materials/Minerals (%) | $Fe_2O_3$ | $Al_2O_3$ | $SiO_2$ | $TiO_2$ | $Na_2O$ | CaO | LOI | Average PCS | $SiO_2/Al_2O_3$ |
|---|---|---|---|---|---|---|---|---|---|---|
| 1 | Red mud [15] | 41.43 | 20.61 | 7.36 | 10.28 | 0.43 | 8.92 | 9.91 | 78.32 | 1.63 |
| 2 | Metakaolin [15] | 1.84 | 40.98 | 52.66 | 1.42 | 0.56 | 0.18 | ND | 95.66 | |
| 3 | Slag (GGBS) [16] | 0.2–1.6 | 7–12 | 27–38 | ND | ND | 34–43 | ND | 34.2–51.6 | 2.52–3.42 |
| 4 | Rice husk ash [16] | 0.26 | 0.39 | 94.95 | ND | 0.25 | 0.54 | ND | 95.6 | |
| 5 | Slag (BFS) [17] | 1.2 | 9.6 | 32.30 | 2.2 | 0.5 | 38.5 | ND | 43.10 | 5.17 |
| 6 | Zinc mine tailing [17] | 19.23 | 6.40 | 25.15 | 0.09 | 0.23 | 0.83 | 18.68 | 50.78 | |
| 7 | Metakaolin [17] | 1.10 | 43.50 | 52.50 | 1.80 | 0.30 | 0.20 | 1.30 | 97.10 | |
| 8 | Bagasse ash [8] | 3.0 | 5.0 | 65.0 | ND | ND | 9.0 | 17 | 82.00 | 13.00 |
| 9 | SFCC [18] | 0.91 | 41.57 | 48.09 | 0.85 | ND | 0.22 | 2.19 | 90.79 | 1.15 |
| 10 | GGBS [19] | 0.30 | 8.51 | 40.30 | ND | ND | 37.01 | ND | 86.12 | 4.70 |
| 11 | Bagasse ash [20] | 3.0 | 5.0 | 65.00 | ND | ND | 9.0 | 17 | 73 | 13.0 |

WTR: Water Treatment Residue; SCBA: Sugar Cane Bagasse Ash; PCS (Pozzolanic Content Summation): $SiO_2 + Al_2O_3 + Fe_2O_3$; SFCC: Spent Fluid Catalytic Cracking Catalyst.

Another study on pozzolanic mineral-rich industrial wastes was carried out by Sundeep et al. [16], to utilize GGBS (ground granulate blast furnace slag) and RHA (rice husk ash) as a supplementary binding material to develop geopolymer concrete. The study was conducted to determine the technical and mineralogical feasibility of pozzolanic-rich mineral waste GGBS and RHA, to reduce the dependency on fly ash for geopolymers. Different material compositions with different ratio of $SiO_2/Al_2O_3$ were used, from 2.58 to 3.42 in GGBS, varying percentage compositions with fly ash, i.e., 0–100% replacement. Meanwhile, 2.58 to 3.57 $SiO_2/Al_2O_3$ was analysed from 0–20% replacement of RHA with fly ash. The chemical compositions of GGBS and RHA are shown in Table 1. It was observed that the highest $SiO_2/Al_2O_3$ ratio at 3.42 for 100% GGBS replacement to fly ash does not show promising results, compared to the minimum ratio of 2.52. Similarly, it was observed that the replacement of fly ash up to 20% could be possible with RHA because an excess of alumina-silicates beyond the ratio of 2.5 disrupts the geopolymer process and creates an unreacted geopolymer mass which decreases the strength. Hence, it can be noted that the exact optimization of the activator ratio is required for the development of predefined strength-specific geopolymer concrete.

A different origin with multi treated waste as a raw material for geopolymer synthesis was conducted by Paiva et al. [17] with zinc mine tailing, metakaolin and slag-based geopolymer synthesis, in which they found that the metakaolin, having an amorphous form of composition and a sufficiently considerable range of PCS, shows that the synergistic proportions of metakaolin with slag can produce geopolymer composition with more than 20 MPa strength. The average materials $SiO_2/Al_2O_3$ ratio was observed as 5.1 & 7 which may have a strength-effective effect due to materials stabilizing properties in geopolymer reaction. It was also evident that the curing and setting properties of the mentioned geopolymer compositions could be well controlled, due to the highly stabilized geopolymer properties of raw materials. However, metakaolin with mine tailing based geopolymer compositions showed up to 22 MPa in (50:50) mass contribution, rather than 15.4 MPa in (50:50) with slag.

The use of the biomass-based materials due to rich pozzolanic mineral components like $SiO_2$ have stretched the material utilization from RHA and bagasse ash (BA). As discussed from Pryscila Andreao et al. [8] the study has investigated on the mineral contents of bagasse ash for the replacement of cement. Thermo-mechanical treatment was provided to bagasse ash to achieve the desired chemical composite, similar to Portland cement. As a result of the investigation, they concluded that recalcination and grinding of ash can subsequently increase the strength performance of concrete. As per the investigation, the activated amorphous silica increased the reactivity due to calcination and continuous grinding. The final content of silica in treated bagasse ash was observed at around 70–75%, which played an important role in increasing the pozzolanic reaction efficiency.

Trochez et al. [18] assessed the precursor-based methodology to utilize spent fluid catalytic cracking catalysts (SFCC) for geopolymer synthesis. The study was carried to understand the trend for different changing parameters in $SiO_2/Al_2O_3$ and $Na_2O/SiO_2$ molar ratios. It was reported that the formation of alumino-silicates was associated with the formation of zeolites. Optimized compressive strength of up to 67 MPa was observed for $SiO_2/Al_2O_3$ and $Na_2O/SiO_2$ ratios of 2.4 and 0.25, respectively. Thus, this investigation could set a new criterion for spent liquid wastes as a precursor for geopolymer liquid that can reduce the commercial consumption of sodium hydroxide (NaOH).

In some studies, the ratio criteria for geopolymer compositions were utilized with $Na_2O$ instead of $Al_2O_3$. An investigation was conducted by Sisol et al. [19] on the $SiO_2/Na_2O$ ratio and the content of $Na_2O$ as an alkali activated precursor to develop hardened blast furnace slag-based geopolymers. The varying ratios of $SiO_2/Na_2O$ at 0, 0.25, 0.5, 1.0, 1.2 and 1.4 were adopted in a series of investigations, to assess the mechanical strength of compositions. The highest seven days' compressive strength was observed as 44.2 at 1.2 ratio factor, whereas 28 days' compressive strength was observed as 52.7 MPa. A similar study was adopted by Ruzkon and Chindaprasirt [20] for different alkali ratios, to assess the feasibility of bagasse ash for low-strength porous geopolymer mortar synthesis. The constant 10 M NaOH to sodium silicate ratio was used as 2.5 at 65 °C accelerated curing condition, which proficiently produced 11.6–15.8 MPa strength geopolymer mortar. The chemical compounds of the above-cited industrial wastes are shown in Table 1.

Hence, from the cited investigations, it is found that pozzolanic contents like $SiO_2$, $Al_2O_3$, $Fe_2O_3$ and CaO play an important role in making geopolymer mineral bonded alumino-silicates. As per the above-cited work of Sundeep et al. [16], the $SiO_2/Al_2O_3$ ratio of approximately 2.5 is identified as a best possible cost-effective alkali activator ratio which could be maintained during the activator optimizations. Ratio of $SiO_2/Al_2O_3$ more than 2.5 could hamper the geopolymer reaction due to unreacted surface efflorescence effect. The equally balanced amount of pozzolanic minerals in the waste benefits the geopolymer reaction to form stabilized concrete. Geopolymer raw materials, like red mud, fly ash and slag/GGBS, can be considered a reference geopolymer raw material, since their mineralogical composition shows balanced contents of pozzolanic compounds, i.e., $Al_2O_3$, $SiO_2$ $Fe_2O_3$&CaO, which undergo geopolymerization, to form oxides of calcium and alumino-silicates of activators with the raw material. Hence, high strength (70 MPa) development in geopolymers has been identified in slag-based materials, whereas materials like red mud, fly ash, RHA and bagasse ash need geopolymer precursors for their reaction. Hence, wastes like red mud, fly ash, slag, rice husk ash, mine tailings, metakaolin, spent liquid waste and bagasse ash can also be utilized to develop low strength (10 MPa) to high strength (70 MPa) geopolymers.

## 3. Surface Morphological Characterization through Scanning Electron Microscopy (SEM) for Engineered Geopolymer Concrete

The microstructural analysis provides evidence of the physical microstructure of geopolymer. This has been conducted by Koshy et al. [21] on aluminium industrial waste, i.e., red mud and thermal power plant coal burnt i.e., fly ash, for micro structural geopolymers' characteristics, within varying ages of 7 days, 14 days, 21 days and 28 days. It is reported that, after 7 days of curing, fly ash showed a rounded but loose pattern, due to less

reactivity of particles with alumino-silicate gel around them. It was reported that red mud itself contains appreciable amounts of soda, which accelerates the geopolymer alkaline-dissolving reaction with silica and alumina particles. After the complete consumption of reactive soda, the dissolution of fly ash stops. The morphological interface and bond of fly ash with alumino-silicate gel showed a weak zone of precipitation, which leads to loosening of fly ash particles and causes cracking to start, due to early hydration; the fly ash particles get carved, due to the strong alkaline environment.

Al Bakri et al. [22] studied power plant coal burnt waste, i.e., fly ash, as a mechanically activated precursor, to study the effect of fineness on the compressive strength of fly ash based nano concrete. The fly ash derived from coal combustion was mechanically nano processed by means of a high-energy planetary ball mill (HEPBM). The particles' morphological behaviour was kept under scanning observation, for unprocessed fly ash and processed fly ash, for 2 h, 4 h and 6 h. The geopolymer process was adopted for the development of alumino-silicate gels, to strengthen the concrete matrix. A time-based strength study from the first day to the seventh day was investigated to compare the compressive strength of geopolymer concrete due to changes in the curing period. As a result of the investigation, it was noted that the fineness of the pozzolanic material plays a vital role in the development of alumino-silicate gels, which strongly influences the development of compressive strength. The scanning electron microscopy (SEM) studies of fly ash processed at nano level at different intervals are explained above.

Hence, based on the above studies, it can be predicted that the appropriate addition of alkali is an important requirement to prevent early hydration and extra etching of the feed materials. In order to maintain the morphological characteristics of the feed raw materials, it is necessary to develop algorithm-based alkali optimizing models to calculate exact amount of alkali activators' requirement for geopolymer concrete mixes. The contribution of fineness can also play a vital role in intermolecular bindings between the particles, due to mechanical activation characteristics that enhance the mechanical strength of concrete mixes.

## 4. Geopolymer Synthesis

Geopolymer synthesis can be processed and synthesized by both dry and wet systems. Bayuaji et al. [23] reported two different methods for the synthesis of fly ash based geopolymers in dry and wet forms. Similar to cement concrete mix designs, they considered geopolymer as a cement, and a water to binder ratio was decided as per molar concentration or dry density of activator compounds, i.e., sodium hydroxide (NaOH) and sodium silicate ($Na_2SiO_3$). In their study, they found that dry geopolymer has an easy method of geopolymer synthesis. The dry geopolymer could have advantages over wet geopolymer systems, due to its ease of manufacturing and in situ applications. Abdel-Gawwad and Abo-El-Enein [24] successfully demonstrated dry geopolymer synthesis with their novel modifications to prevent hydrophilicity of the dry geopolymers. They adopted the use of water-absorbing compounds to prevent lump formation in dry geopolymers. This study shows the promising incorporation of calcium carbonate ($CaCO_3$), sodium carbonate ($Na_2SiO_3$) and prissonite (P), followed by mechanical milling to stabilize the dry activators. The steel industrial slag, i.e., granulated blast furnace slag (GBFS), was used as an activated pozzolanic material, to develop new-age dry geopolymer cement. Chandra Padmakar and Chandra Kumar [25] worked on the synthesis of geopolymers, using steel industrial waste slag through liquid-based geopolymer synthesis. The main contents of the liquid activators used were a 10 M sodium hydroxide solution and a solution of sodium silicate solution. The geopolymer mix was formed, containing GGBS and metakaolin in place of cement, since the chemical constituents of granulated blast furnace slag (GGBS) and metakaolin show an appreciable amount of alumina and silica-based geopolymer precursors. These geopolymer precursors can be utilized to provide alkali-activation to the geopolymer mixings. Kiran Kumar and Gopala Krishna Sastry [26] investigated fly ash-based geopolymer concrete, using a liquid activation process.

The geopolymers can also be synthesized with liquid activators, using sodium hydroxide and sodium silicate at ambient temperature. From the above-cited reference, it could be noted that the synthesis of geopolymer can be adopted by both wet and dry process routes. Moreover, dry geopolymer synthesis has better workability, compared to wet geopolymer synthesis, since the wet geopolymer synthesis involves a slow process from raw material preparation to the decision on the molar concentration of activator solutions. The strength-specific study was conducted by Gum Sung Ryu et al. [27] on the synthesis of fly ash based wet geopolymers. The study shows the effect of different molar concentration studies with varying sodium hydroxide and silicate ratios, e.g., 6, 9 and 12 M, and also with different percentage variations, e.g., 0:100, 25:75, 50:50, 75:25 and 100:0. The findings show that synergic use of two activators, i.e., sodium hydroxide and sodium silicate, plays a vital role in the development of compressive strength. It was reported that the increasing molar ratio of 12 M sodium hydroxide with 50% sodium silicate solution was limited to 47 MPa strength, whereas 9 M sodium hydroxide solution reflected a result close to 47 MPa. Meanwhile, the percentage variations of sodium hydroxide to sodium silicate (NaOH:$Na_2SiO_3$%) show the compressive strength of 47 MPa at the optimized ratio of 50:50. This illustrates that the geopolymer strength can be altered, by variations in the molar ratio of activators.

An effective temperature treatment for geopolymers is evident through the work of Nan Ye et al. [28]. The studies were aggressively focused on evaluating the effect of temperature treatment on red mud and GBFS-based wet geopolymers. The wet system was adopted to ascertain the moisture requirement for geopolymer formations during temperature studies, to prevent drying shrinkages and cracks. A series of trial runs of temperature effects was carried out from 100 to 800 °C, with progressive evidence of 25 to 50 MPa in strength increment. Furthermore, geopolymer material was also found suitable for fire-resistant construction materials. An added physical material development of thermal-resistant geopolymer was investigated by Cheng and Chiu [29] for the development of fire-resistant geopolymer panels, which were synthesized by a slag-based geopolymer matrix. It is evident that the slag-based geopolymer could withstand extreme temperatures up to 1100 °C and could sustain the post-treated strength till 79 MPa. As an accelerated procedure for geopolymer nanoscale alumino-silicates' dissolution, the lower alkali-water ratio was observed at around 0.026. The contribution of biomass industrial waste for making geopolymer was studied by Zabihi et al. [30], who investigated the effect of rice husk ash, which was added as a $SiO_2$ mineral contributor to the wet geopolymer system. The use of fibres with RHA as a 100% replacement was adopted to develop cement-free concrete. The fibrous properties and pozzolanic properties of RHA contributed to a strength development of 65 MPa, with 63% $CO_2$ less energy-intensive outputs. Additionally, the contribution of biomass, e.g., bagasse ash, to geopolymer synthesis satisfied the making of medium-strength geopolymers, as reported by Rukzon and Chindaprasirt [21]. The delivered synthesis comprised of low, porous, medium-strength, complete waste and accelerated the cured geopolymer route to optimize the desired low-strength-based geopolymers.

As per the cited reference, the suitability of the geopolymer synthesis system could be well adopted in both wet and dry processes, since the alkalis have good ability to be converted into a dry powdered form and also show a solute nature to make aqueous activators. The important synthesis parameters with their findings are stipulated in Table 2.

**Table 2.** Different geopolymer concretes from different industrial wastes.

| SI | Reference | Synthesis | Waste | Activator Compound (NaOH: $Na_2SiO_3$) | Optimized $SiO_2/Al_2O_3$ Ratio |
|----|-----------|-----------|-------|----------------------------------------|----------------------------------|
| 1 | Ridho Bayuaji et al. [23] | Wet | GBFS | (1:2.92 M) | ND |
| 2 | Abdel-Gawwad and Abo-El-Enein [24] | Dry | GBFS | Dry NaOH | 2.87 |
| 3 | Chandra Padmakar and Sarath Chandra Kumar [25] | Wet | GGBS & metakaolin | (1:2.5) | 2.61 |

**Table 2.** *Cont.*

| SI | Reference | Synthesis | Waste | Activator Compound (NaOH: Na$_2$SiO$_3$) | Optimized SiO$_2$/Al$_2$O$_3$ Ratio |
|---|---|---|---|---|---|
| 4 | Kiran Kumar and Gopala Krishna Santry [26] | Wet | Fly ash, GGBS and nano silica | (1:2) (weight/weight) | 3.6 |
| 5 | Ryu et al. [27] | Wet | Fly ash | NaOH = 6 M, 9 M & 12 M & Constant NaOH 9 M for fly ash (100, 75, 50) | 4.13 |
| 6 | Nan Ye et al. [28] | Wet | Red mud & GBFS | (1:6.9) | 1.7 |
| 7 | Cheng and Chiu [29] | Wet | GBFS & metakaolin | KOH (5–10 M) | 2.37 |
| 8 | Zabihi and Mohseni [30] | Wet | Rice husk ash | Sodium hydroxide 10 M | 3 |
| 9 | Sumrerng Rukzon and Prinya Chindaprasirt [21] | Wet | Bagasse ash | (10:15) | 13 |

## 5. Mechanical Properties of Geopolymer Cement/Concrete

The mechanical properties of geopolymer concrete made from industrial waste are given in Table 3. Tran Viet Hung et al. [31] carried out a study evaluating the mechanical properties, in which they investigated the strength behaviours of fly ash based geopolymer concrete for compression, flexural, elastic modulus and tensile loadings. The mechanical properties of flexural strength and the characteristics of bending tensile strength were compared with cement concrete compositions. It was concluded that the tensile strength of geopolymer concrete shows promising behaviour, compared to that of cement concrete, which directly indicates that the cracks in the geopolymer concrete are less than normal cement concrete. Another study on the synergic utilization of fly ash and slag was performed by Sofi et al. [32], for the synthesis of fly ash slag-based geopolymer concrete, in which they compared the geopolymer fly ash-based concrete with conventional, ordinary Portland cement concrete. The observed mechanical characteristics of geopolymer concrete were seen to be similar to ordinary Portland cement (OPC) in the case of compressive strength and standard deviations. However, split tensile strength and flexural strength were observed to be quite differentiated, with a 2 MPa strength range. The modulus of elasticity was reported to be more promising than OPC concrete. The density of geopolymer concrete was in the range of 2147–2408 Kg/m$^3$, whereas the OPC concrete showed a density range of 2300–2600 Kg/m$^3$. Compressive strength was achieved in the range of 47–56.5 MPa and showed increasing strength from 10–15 MPa strength in between 7 and 28 days. This may be due to evidence of polymerization reaction even after 7 days of curing, up to 28 days at ambient room temperature conditions. The other mechanical properties of this study are shown in Table 3. The geopolymer processes also have room for analysis on a singular liquid activator medium. The study of Fernandez-Jimenez et al. [33] shows similar research on the engineering properties of fly ash based geopolymer concrete. The fly ash based geopolymer synthesis was bifurcated in two liquid medium systems, via sodium hydroxide based and sodium silicate-based aqueous activators. The sodium hydroxide, which was denoted as the N activator, was kept at a constant molar concentration of 8 M, whereas the W activator, i.e., sodium silicate, was kept at a molar concentration of 12.5 M, while the homogenous mixture of both the mixtures was kept at a 15:85 ratio. The N and W based geopolymer syntheses showed promising results in compressive strength, which was 34 and 43.5 MPa, respectively, compared to OPC concrete's 42 MPa. The accelerated curing showed a rise in the compressive strength up to 60 MPa at 672 h, compared to OPC concrete at approximately 30 MPa at a similar time of curing. Regarding flexural strength, the trend

shows a linear regression from 4.62 MPa to 8.80 MPa from 20 h to 90 days. The other mechanical strength properties of this literature are stipulated in Table 3.

**Table 3.** Mechanical properties from several studies by researchers.

| Reference | Mechanical Properties | | | | | | | | | |
|---|---|---|---|---|---|---|---|---|---|---|
| | Activator Ratio | Density | CS (MPa) | STS (MPa) | Slump (mm) | FS (MPa) | MOE (GPa) | A/B Ratio | Temp. & Time | Waste |
| Viet Hung et al. [31] | 2.5 | ND | 52.07 | 33.65 | ND | 7.31 | 33.65 | 0.41–0.48 | 60 °C/24 h | Fly ash |
| Sofi et al. [32] | ND | 2147–2408 | 47.0–56.5 | 2.8–4.1 | ND | 4.9–6.2 | 23.0–39.0 | 0.45–0.59 | 23 °C RT | Slag & fly ash |
| Fernandez-Jiminez et al. [33] | 8 &12.5 M | 2400 | 29.0–43.5 | ND | ND | 6.86 | 10.7–18.4 | 0.40&0.55 | 85 °C/20 h | Fly ash |
| Gum Sung Ryu et al. [27] | 6, 9 &12 M | ND | 15.2–47.5 | 1–5 | ND | ND | ND | 0.5 | 60 °C/24 Hrs | Fly ash |
| Gunasekara et al. [34] | 7% (w/w) | 2000–2682 | 31–64 | ND | 150–175 | 3.95–4.95 | 18–27 | ND | 140 °C/15–90 min | Kaolin & fly ash |
| Abbas et al. [35] | 10–16 M NaOH | 1835 | 35.8 | 2.6 | 245 | 5.5 | ND | 0.25 & 0.61 | 60 °C/48 Hrs | Low Ca fly ash |
| Chamundeswari and Ranga Rao [36] | (1:2,2.5,3.0) | 2445 | 8–12 | 0.21–0.71 | ND | 0.2–1.7 | ND | 0.45 | 27 °C RT | Fly ash & metakaolin |
| Gautam et al. [37] | 3, 4, 5 & 6 M NaOH | 2322 | 50–70 | 2.78–6.1 | ND | 3.3–5.9 | ND | 0.46–0.62 | Ambient RT | GBFS & fly ash |
| Madheswaran et al. [38] | 3, 5 & 7 M (NaOH & Na$_2$SiO$_3$) | ND | 25–60 | 3.96–5.3 | 75–100 | ND | 13.5–14.14 | 0.65 | Ambient RT | GGBS |
| Singh et al. [39] | 6, 8, 10 & 12 M NaOH | 2400 | 8.8–56 | 2.4–2.8 | 120 | 4.75–7.62 | ND | ND | Ambient RT | Red mud, fly ash & GGBS |
| Kishore K et al. [40] | 10 M NaOH | 2400 | 20–65 | 1.0–6.0 | ND | 1.0–6.0 | ND | 0.4 | Ambient RT | RHA & GGBS |
| Jaya Kumar et al. [41] | 12 M NaOH | ND | 11–38 | 3.2–10.5 | ND | ND | ND | 0.5 | 60 °C/24 Hrs | Fly ash & bagasse ash |

CS: Compressive Strength; STS: Split Tensile Strength; FS: Flexural Strength; MOE: Modulus of Elasticity; ND: Not Defined; M: Molar; A/B: Activator to Binder Ratio.

The effect of different molar concentrations of geopolymer synthesis was investigated in a study conducted by Gum Sung Ryu et al. [27] on fly ash based geopolymer concrete. Varying concentrations of sodium hydroxide were adopted, 6 M, 9 M, and 12 M, to examine concrete reactivity. Additionally, in order to optimize the raw material reactivity, five different percentage composition doses were investigated as (NaOH:Na$_2$SiO$_3$ = 0:100; 25:75; 50:50; 75:25 and 100:0) ratios for an activator medium. The maximum compressive strength achieved at 28 days was 44.8 MPa at 12 M NaOH with 50% sodium silicate activator medium. Additionally, the maximum split tensile strength was observed as an increasing trend of 1–5 MPa with increasing molar concentration ratio. The curing condition was adopted as accelerated type, having a 60 °C curing medium for 24 h. Geopolymers have a wide variety of syntheses, not only in cement and concrete but also in the synthesis of concrete ingredients like aggregates. Gunasekara et al. [34] conducted an extensive study on the synthesis of geopolymer aggregate concrete, in which the density of the geopolymer concrete was adopted in varying ranges from 2000–2682 Kg/m$^3$. Temperature-cured aggregates at a maximum of up to 140 °C were adopted to ensure complete stabilization of the geopolymer matrix. As a geopolymer aggregate synthesis, the concrete was tested at different durations, viz. 7, 28 and 90 days. The increasing strength trend was from 26 to 37 MPa with varying density from 2000 to 2200 Kg/m$^3$. The varied testing conditions of dry and wet were also examined under compression testing, which showed 31 MPa in wet testing and 64 MPa in accelerated 140 °C cured testing.

Research advances for the requirement for high-strength geopolymer can also be achieved, as in the study of Abbas et al. [35], in which they worked on the development of high-strength lightweight geopolymer concrete, with strength conforming to around 36 MPa. Low calcium fly ash, along with alkali activators, were used as geopolymer precursors. The authors succeeded in the synthesis of geopolymer with a low-density concrete mix up to 1800 kg/m$^3$. The flexural strength was observed to increase by up to 3% from 28 to 56 days. The authors stated that this lightweight geopolymer concrete can be used as an insulated high-strength geopolymer that shows comparably less thermal coefficient than normal concrete. Chamundeswari and Ranga Rao [36] conducted research on the effect of molar concentrations on geopolymer mix designs for different intervals of time with different molar concentration ratios. The study shows that the compressive strength increases with increasing molar concentration of alkali activators.

Through this study, it was recorded that the compressive strength of 1:3 (NaOH:Na$_2$SiO$_3$) shows an increase in strength, compared to the 1:2 molar concentration ratio. As discussed above, not only are light-weight high-strength geopolymers being studied; in their study, Gautam et al. [37] show that the high strength of geopolymer concrete can be developed up to 71 MPa with the use of slag and fly ash as a raw material. They investigated the development of high-strength geopolymers with different molar concentration studies, which varied from 3 M to 6 M of sodium hydroxide activator. It was reported in their results that the strength developed at 28 days was around 50 MPa for a 3 M activator solution, around 65 MPa for a 4 M activator solution and around 71 MPa for a 6 M activator solution. The authors clearly mentioned that the increase in molar concentration can alter the compressive strength of the geopolymer mixes. Studies in which the strength is extended up to 52 MPa with optimizing molar concentration can be witnessed in the work of Madheswaran et al. [38]. They studied the effect of varying molar concentration on different GBFS-based geopolymer grades. The study was conducted to obtain the possible trends in strength changes due to the molar concentration changes of alkali activators. From the findings, it was reported that a 7 M sodium hydroxide alkali activator shows promising behaviour, compared to that of a 3 M sodium hydroxide solution. The compressive strength of the 7 M solution attained a maximum of 52 MPa, compared to 3 M at 16 MPa. However, the utilization of red mud as a pozzolanic binder with a synergic approach with other industrial waste like GGBS and fly ash (FA) have shown promising strength results: up to 56 MPa at 28 days at ambient room temperature, as investigated by Singh et al. [39]. The optimized material replacement for red mud in geopolymer concrete was suggested as 30% with the combination of GGBS and FA. The optimized strength parameters are stipulated in Table 3. It was reported that the percentage of red mud replacement and different molar concentrations affect the strength properties of the geopolymer concrete. As silica-accelerated agents, in the form of synergic use of biomass waste, viz. RHA and bagasse ash, can nurture the pozzolanic reactivity of high-strength waste materials, such as GGBS and fly ash, they can effectively consume biomass waste, to obtain value additions. The investigations performed by Kishore K et al. [40] and Jaya Kumar et al. [41] show that strength achievements of up to 65 MPa in the synergic use of RHA and GGBS could be possible at ambient room temperature at constant 10 M NaOH concentration, whereas the accelerated curing at 60 °C can make a partial utilization of bagasse ash with fly ash to develop a low-strength geopolymer up to 38 MPa. However, the high molar concentrations (12 M) of the bagasse and fly ash based geopolymers are optimized by the studies, but the strength development confirmations were identified by this study.

According to the literature discussed above, mechanical strength parameters like compressive strength, flexural strength and tensile strength could be achieved similarly to conventional concrete mix designs. It can be seen that the compressive strength of geopolymer concrete reached with fly ash 15–50 MPa, GGBS 25–70 MPa, red mud 8.8–56 MPa, RHA 20–65 and bagasse ash 11–38 MPa. Furthermore, the effect of molar concentration on the strength of geopolymer concrete is an observed factor since the dissolution rate and efficiency improve when the molar concentration of the alkali activator increases. It is observed that molar concentration ranges from 6 to 10 M of NaOH and sodium silicate are suitable for complete geopolymerization reaction for all industrial wastes. High molar concentrations show better compressive strength than their lesser molar concentration counterparts. The molar concentration also contributes to the dissolution of minerals like silica and alumina that contribute to the formation of alumino-silicate gels later; as the curing period increases, it strengthens the concrete mix. As shown in Table 3, it is observed that the effect of temperature on geopolymers varies from material to material. Slag, as a raw material, shows an equal mineral content of SiO$_2$ ad CaO; it does not need accelerated temperature curing and can be polymerized to form good-strength geopolymers, while an excessive amount of SiO$_2$ in fly ash needs accelerated temperature curing because the excess of silica content leads to slow drying conditions, as shown in Table 3, because of

its high content of SiO$_2$. Geopolymer synthesis can be done at varying temperatures, from 23 °C to 140 °C. It has been observed that fly ash 15–50 MPa, GGBS 25–70 MPa, red mud 8.8–56 MPa, RHA 20–65, and bagasse ash 11–38 MPa geopolymer concrete satisfies the strength criteria in concrete applications. Among this, slag-based geopolymer with fly ash, red mud, and biomass waste can be used to develop high-strength competitive geopolymers in comparison to normal cement concrete. Moreover, as discussed by Viet Hung et al. [31], porosity can be controlled to decrease the effect of weathering actions, indirectly reducing the compressive strength of concrete.

The comparative studies of geopolymer-based compositions of different industrial wastes show different mechanical strengths. Figure 2 represents the bar chart-based comparison of fly ash and fly ash with other waste-based geopolymers versus the strengths for compression, split tensile and flexural loadings. As described by Viet Hung et al. [31], singular fly ash based geopolymer concrete shows permissible strength limits up to 52 MPa compressive strength, 5 MPa split tensile and 7.31 MPa flexural strength, compared to normal cement concrete, whereas the synergic use of GGBS and fly ash showed more promising behaviour, compared to fly ash based geopolymer concrete. The strength development is directly correlated with the material's mineral composition, which is shown in Table 1. GGBS and fly ash contain a considerable amount of SiO$_2$, CaO and Al$_2$O$_3$. Similarly, kaolin and metakaolin have a similar minerology of pozzolanic contents and, hence, high strength, i.e., it could be possible to synthesize >50 MPa for fly ash based geopolymer compositions. The additive use of fly ash and GGBS can possibly attain the synergic use of pozzolanic-rich aluminium industrial waste, i.e., red mud, to obtain a high-strength performance factor for its bulk utilization. In addition, biomass industrial waste, either rice husk ash or bagasse ash, has a scope of utilization for geopolymer synthesis, as a supplementary cementitious material, to also develop low-strength geopolymers.

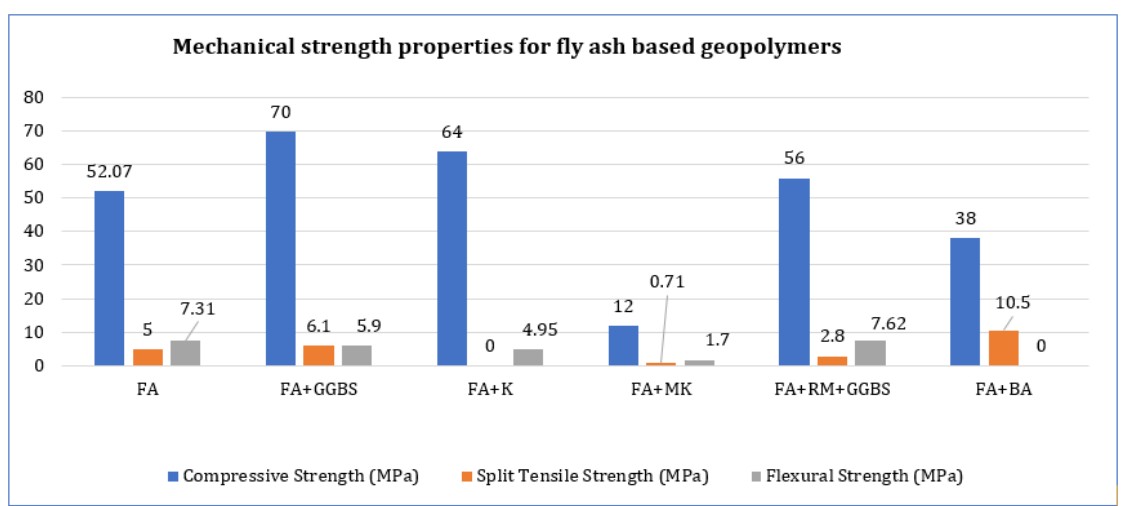

Remarks: *FA- Fly Ash; GGBS – Ground Granulated Blast Furnace Slag; K – Kaolin; MK – Metakaolin; BA- Bagasse Ash*

**Figure 2.** Graph showing synergic fly ash based geopolymers' compositions versus compressive strength.

The mineralogical and stabilized morphological characteristics of GGBS, i.e., slag, have evidence of high-strength-based geopolymer, as shown in Figure 3. The mechanical strength behaviour of GGBS-based geopolymers confirms good structural stability under compression, tensile and flexural behaviour. GGBS shows high strength (>50 MPa) compressive strength, (>5 MPa) tensile and (>5 MPa) flexural loadings, not only as a singular waste-based geopolymer, but also demonstrates a similar strength behaviour with rice husk ash and fly ash. However, although high strength (>50 MPa) is possible with red mud, for compressive strength, it has less (<2 MPa) tensile and flexural load stability.

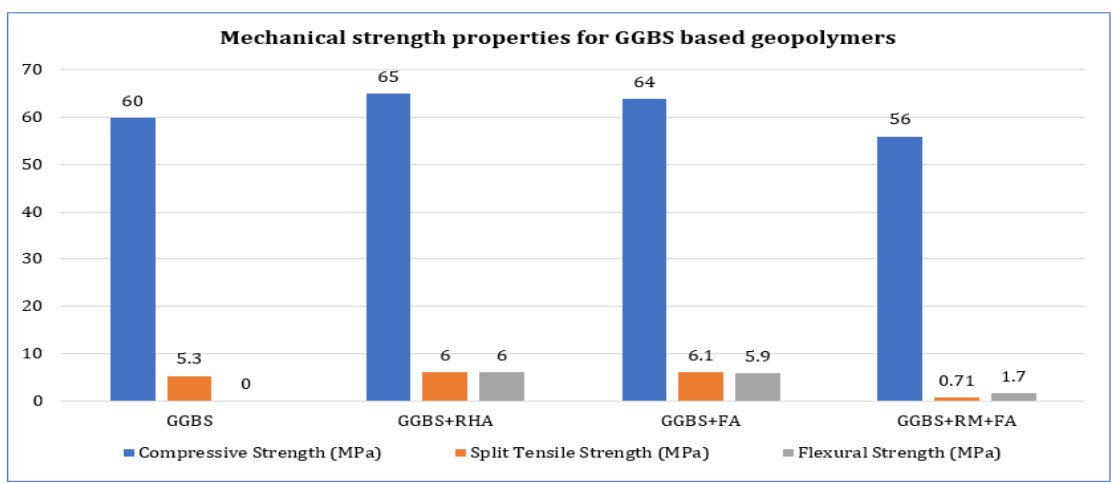

*Remarks: FA- Fly Ash; GGBS – Ground Granulated Blast Furnace Slag; K – Kaolin; MK – Metakaolin; BA- Bagasse Ash*

**Figure 3.** Graph showing synergic GGBS-based geopolymers' compositions versus compressive strength.

Geopolymer is remarkably influenced by accelerated temperature curing, since the geopolymer reaction has an effect on temperature conditions, as shown in Figure 4. Geopolymer has huge scope for temperature-based synthesis, as it is been observed in the table that the geopolymer synthesis carried out at room temperature for GGBS-fly ash and red mud-fly ash-GGBS based geopolymers showed compressive strength greater than 50 MPa, except for fly ash-metakaolin based geopolymer. However, the accelerated temperatures at 60, 85 and 140 °C do cause effective increments in the strength of geopolymer concretes made by fly ash and low-strength based industrial waste like kaolin and bagasse ash. Hence, from this study, it can be noted that geopolymer concrete can be prepared for high-strength targets with high pozzolanic mineral valued waste materials like GGBS, fly ash and red mud, whereas thermally cured low pozzolanic mineral waste geopolymer could be developed to high strength by thermal curing.

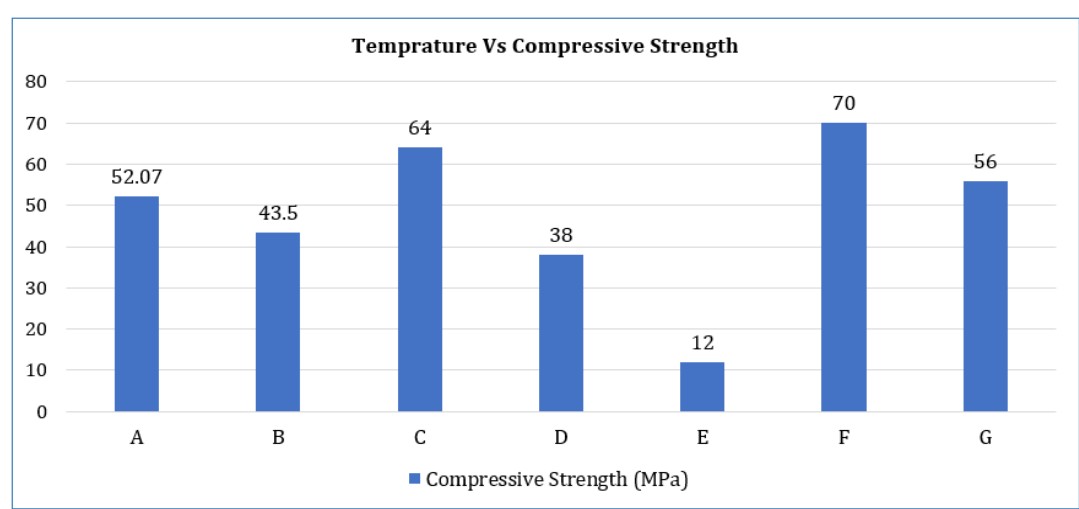

**A** -FA 60°C/24 Hrs **| B**- FA 85°C/20 Hrs**| C**- FA+K -140°C/1.5 Hr **| D**- FA+BA-60°C/24 Hrs**| E**- FA+MK- RT**| F**- GGBS+FA – RT **| G**- RM+FA+GGBS-RT

**Figure 4.** Graph showing temperature vs compressive strength.

The activator to binder (A/B) ratio has a notable effect on the strength of geopolymers. From Figure 5, it has been observed that compositions with the A/B ratio, with a range of 0.4 to 0.65, have shown promising compressive strengths greater than 50 MPa. How-

ever, the lowest A/B ratio was observed as 0.4 in the case of GGBS + RHA composition. The reason for the remarkable compressive strength of up to 65 MPa for the GGBS+RHA composition could be assumed to be the development of silicates of calcium and of alumina, since the mineralogical contents of GGBS and RHA show rich pozzolanic minerals like $SiO_2$, $Al_2O_3$ and CaO. In contrast, the highest A/B ratio, i.e., 0.59 for GGBS+FA composition, could have excess pozzolanic accumulation during the process, which may lead to the development of an unreacted mass that reduces the compressive strength of geopolymers.

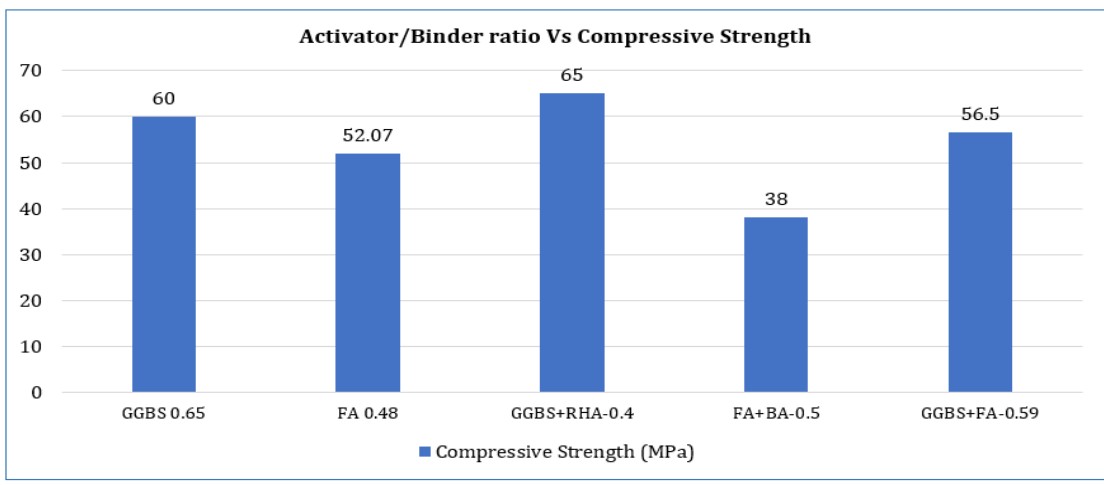

**GGBS 0.65 –** GGBS geopolymer with 0.65 Activator to Binder ratio| **FA 0.48 –** Fly Ash geopolymer with 0.48 Activator to Binder ratio | **GGBS+RHA-0.4 –** GGBS and RHA based geopolymer with 0.4 Activator to Binder ratio | **FA+BA 0.5 –** Fly Ash and Bagasse ash based geopolymer with 0.5 Activator to Binder ratio | **GGBS+FA-0.59 –** GGBS and fly ash based geopolymer with 0.59 Activator to Binder ratio

**Figure 5.** Graph showing activator/binder ratio vs compressive strength. Remarks: FA—Fly Ash; GGBS—Ground Granulated Blast Furnace Slag; K—Kaolin; MK—Metakaolin; BA—Bagasse Ash; RM—Red Mud; RHA—Rice Husk Ash.

## 6. Durability of Geopolymer Concrete

### 6.1. Alkali-Silica Reaction

Investigation of alkali-silica reaction provides both the mineral's feasibility to react and the amount of degradation that could hamper the geopolymer matrix due to unreacted or free soda state. According to the study reported by Patil and Allouche [42], it has been seen that, in geopolymer reaction, excess alkali of alumino-silicates is susceptible to reacting with the silica of reactive aggregates, which distorts the bonds between the pozzolanic ingredients. The permissible threshold expansion was studied as per ASTM C1260 for fly ash based geopolymer concrete in comparison to OPC. However, it is reported that geopolymer pastes using different reactive aggregates show less expansion, compared to normal concrete. Further, in the case of slag-based geopolymers, Fernandez-Jiminez and Puertas [43] investigated surface morphological studies on alkali-activated granulated blast furnace slag, in which they observed that the geopolymer compositions did not show surface cracking, and the expansion rate was supposed to be observed to be slow, compared to OPC concrete. It is reported that the surface expansion in slag-based geopolymer concrete was observed due to the presence of oxides of calcium, which develop the hydrates of sodium calcium silicate.

However, the work by Garcia-Lodeiro et al. [44] found a contradiction on the point that there was no significant expansion of surface cracking in fly ash based geopolymer mortars. The authors stressed the slow hydration and slow zeolite reaction, concluding that the presence of minerals usually occurs in gaps in the pozzolanic matrix, which, hence, do not exert cracking pressure on the surface of the geopolymer matrix. Alkali-activated red mud geopolymer concrete was studied by Ribeiro et al. [45], who identified the ef-

ficiency of alkali-silica reaction in the geopolymer matrix, due to the presence of a high concentration of alkali in the raw materials' mineral compositions that could bring about a positive rheological change. Since the alkali-silica reaction mechanism creates surface and sub-surface expansion, a study by Ha Thanh Le et al. [46] investigated rice husk and cement paste studies, to find the possible raw materials to mitigate the alkali-silica reaction expansion. As a concluding remark to the study, it was suggested that using rice husk ash with fine ground particles would show complete reaction efficiency. The oxide composition of calcium silicate and hydrates reacts with the sodium and potassium of the minerals and with the siliceous minerals of RHA. However, the study of bagasse ash conducted by Ramjan et al. [47] shows that the finer the particle size of the raw materials, the greater the alkali–silica reactivity, since the fineness of the particles increases the mineral reactivity and makes it more amorphous; it was observed that even the high loss of ignition of the bagasse ash does not alter the alkali–silica reactivity. Thus, the authors encourage the use of bagasse ash as a supplementary binder.

From the above-cited reference, it is evident that the possibility of alkali–silica reaction has a higher chance in calcium-based geopolymers (i.e., since this reaction forms hydroxides of calcium, which have a deteriorating and surface-expanding nature). Hence, the development of sodium-based alumino-silicates should be the priority for the formation of sodium-based geopolymers. The alkali-activated geopolymer slag, red mud and fly ash based geopolymer binders are found to be suitable, due to the mineral-stabilizing properties in alkali–silica reaction because of their multi mineral compositions, whereas the biomass ash, like rice husk and bagasse ash, needs more fineness treatments, which could indirectly hamper the cost benefits of geopolymers.

*6.2. Effect of Acid Attack*

The degree of deterioration of concrete also depends upon the acid exposure conditions and the period of exposure. Davidovits et al. [48] stated that metakaolin-made geopolymers show up to 7% mass loss when immersed in 5% sulphuric acid [$H_2SO_4$] solution for a period of 30 days. It is also evident that the sustainability of fly ash based geopolymer morphology could be retained for more than three months when exposed to nitric acid [$HNO_3$] liquid suspension. Temuujin et al. [49] investigated whether the acid and alkali resistivity of fly ash based geopolymers is due to their mineralogical contents, which strongly resist and neutralize the effect of acidity, due to chemical equilibrium. The complete and high solubility of pozzolanic minerals like alumina (Al), silica ($SiO_2$) and ferrous oxides ($Fe_2O_3$) enables the geopolymer matrix to be more stable, due to the complete reaction mechanism. The acid resistivity of slag-based geopolymer concrete was studied by Ozcan and Burhan Karakoc [50]. In this research, a 5% aqueous acidic medium was investigated, to study the acid resistivity of slag-based geopolymer concrete. The acidic suspensions of phosphoric acid ($H_2PO_4$), hydrochloric acid (HCl) and hydrofluoric acid (HF) were used for 12 weeks, to see the resistivity of the slag-based geopolymer concrete. It was noted that visible surface degradation was greatly observed in the case of 5% $H_2SO_4$ and HF, whereas the effect of alkali attack was less in the case of 5% $H_3PO_4$ and HCl. It was noted that the acidic medium distorts the geopolymer concrete matrix about 3 mm from the concrete surface. Another acid resistance study was performed by Singh et al. [51] on aluminium industrial waste, i.e., red mud under 5% acidic suspension of $H_2SO_4$ and acetic acid ($CH_3COOH$). The cylindrical specimens of red mud based geopolymer concrete were immersed in an acidic medium for 1, 7, 14, 28, 56 and 84 days, in order to monitor the weight loss, visual deformation, strength loss, and physical appearance, as per the procedure specified by ASTM C 267- 01. The red mud based geopolymer paste showed better resistance towards the acidic medium, compared to the cement-based paste. No surface degradation and weight loss of more than 7% was observed in all different compositions. The weight loss of the geopolymer matrix was observed much more in the sulphuric acid than in the acetic acid. However, the authors successfully found the percentage utilization of red mud to be up to 30% of the total mix. Another study of an acid resistance test was carried out on

RHA by Kim et al. [52] in an acidic suspension of 5% $H_2SO_4$ and 5% HCL. The findings were that the RHA-based geopolymer showed good, reduced capillary liquid flow through its material matrix. This resulted in the development of acid-resistant geopolymer concrete, compared to normal cement concrete.

It was evident that the effect of acid causes geopolymers to deteriorate, due to the presence of unreacted alkali in the geopolymer matrix. Hence, it can be assumed that a stable amount of alkali should be incorporated, to prevent the involvement of an excess of free soda. The presence of free soda reacts with the bonded alkali and causes the polymer bond to deteriorate. Insufficient soda/alkali could also hamper the reaction mechanism, since the free unreacted silica and alumina could become involved in the acid reaction, which creates spalling of the concrete.

### 6.3. Effect of Sulphur Attack

Fernandez-Jiminez et al. [53] performed a test to examine the effect of reactive sulphate on geopolymer, which is indicated as a key test to evaluate the sustainability factor of concrete under the influence of sulphur mediums. It was reported that the fly ash based geopolymers did not undergo deterioration when they were subjected to sea water and sodium sulphate solution (4% $Na_2SO_4$). Bakharev [54] studied the mechanical behaviour of class F fly ash based geopolymer concrete. It was observed that the variation in flexural strength, from seven days to three months of exposure, was decreasing in nature. According to the observations, the minimum strength decrement was seen for the geopolymer paste, which was exposed to 5% sodium sulphate and 5% magnesium sulphate solutions. However, the mixtures of these two sulphate solutions reported strength deterioration after five months of exposure. Ismail et al. [55] reported the sulphate attack disrupting the fly ash-based geopolymer bond when it was subjected to liquid suspension of an aqueous magnesium sulphate. It was observed that, after three months of exposure, no considerable physical changes had been noted in the case of sodium sulphate liquid suspension. The reason behind the deterioration of the geopolymer matrix was observed to be due to the formation of calcium sulphate dehydrates, which particularly seem to be damaging in a magnesium sulphate solution. From the above study, it is evident that sulphate attack has degrading properties for geopolymer concrete, but the effect of this sulphate could be neutralized by the addition of some sulphate-stabilizing activators.

### 6.4. Carbonation and Permeability

Bernal et al. [56] investigated the effect of carbonation and permeability for slag-metakaolin based geopolymer concrete under accelerated carbonation, for 28 days at 20 °C. It was observed that the decrease in strength occurs as the carbonation is initiated. The correlation between the pore space and the rate of carbonation was found to be similar for series of different percentages of metakaolin-based geopolymer samples, unlike slag-based geopolymers. It was suggested that the porosity is not the only factor affecting the strength loss for carbonation compounds. There must be a convolution effect, due to alumino-silicate gel chemistry, that defines the undamaged strength after the carbonation process. Olivia and Nikraz [57] reported lower liquid permeability of geopolymer concrete cured at 60 °C for 24 h duration than Ordinary Portland Cement (OPC) based concrete, due to denser material media and less pore interconnectivity. They also concluded that the water to geopolymer solid ratio influences the properties of geopolymer concrete. The carbonation and permeability properties of geopolymer were found to be superior to those of cement-based concrete. The benefits of small pore space ratios of geopolymer and the absorbing properties of carbonate enable the geopolymer to be more sustainable than normal cement concrete.

### 7. Cost and Economics

Regarding the cost concerns of geopolymers, many researchers have been trying to work out how to prepare a cost-effective geopolymer process and parameters. Thaarrini and

Dhivya [58] carried out a study to compare normal conventional cement concrete with geopolymer slag/bottom ash-based concrete, in order to establish the feasibility of geopolymer for commercial applications. In this study, the geopolymer concrete (GPC) was compared with different grades of conventional cement concrete. The research findings show that M30-grade GPC was 1.7% costlier than M30-grade cement concrete, whereas GPC showed good economic benefit in the case of higher grades than M30. The authors worked on a cost analysis comparison of M50-grade GPC and M50-grade cement concrete, in which they found GPC to be 11% cost-effective, compared to conventional concrete. Siming You et al. [59] worked on the techno-economic feasibility of geopolymers, based on fly ash, iron oxide and calcium oxide as a precursor. The research findings show that variations in the materials for geopolymer compositions play a vital role, since the chemical activators, NaOH and Sodium Silicate $Na_2SiO_3$, are the key elements for the cost economics of geopolymers. Another study on red mud (RM)/coal gangue (CG)-based geopolymer concrete was carried out by Junjun Geng [60]. The research reveals the mechanical properties and chemical polymerization route of geopolymers. As a chemically activated precursor for red mud based geopolymer, CG can decrease the molarity of the chemical activator required; i.e., more dilution to the alkali activator is possible, as the raw material itself contains activated chemical compounds like $SiO_2$ and $Al_2O_3$, which reduces the high molar concentration of the alkali activator and, hence, the cost of the geopolymer is reduced. It can be seen that pre-activation of raw materials is vital to make the geopolymer composition more cost-effective. This study opened the way for low-cost geopolymer synthesis and made the utilization of red mud as an ingredient in geopolymer concrete more cost-effective.

Biomass industrial wastes, like RHA and BA, however, are rich in one of the pozzolanic chemical ingredients, i.e., $SiO_2$. Their role in geopolymer reaction starts when the polycondensation of alumino-silicates starts. As a single raw material based geopolymer, biomass ash may have some process restrictions, due to the excess silica, but its contribution to low-cost building materials like bricks can be considered. Poinot et al. [61] investigated a study on RHA-based geopolymer building materials and their cost comparison with conventional fired clay bricks. The cost comparison of energy consumption showed that the geopolymer-based technique requires 1% energy (heat), whereas the fired clay bricks require approximately 58% energy (heat), which shows that the firing process required for 900–1000 °C firing could be reduced, to achieve low carbon-based concrete manufacturing for concrete construction. This research work indicates that the geopolymer bricks can be produced at a similar production cost to that of clay bricks, with a reduced environmental impact, making them viable on the market as an alternative low-cost, low-energy source for building material. Another research work on biomass-based geopolymer was carried out by Syed Nasir Shah et al. [62]. The authors showed a unique way to design a framework to pursue lightweight geopolymer concrete with the use of industrial wastes, including bagasse ash. They elaborated on the studies pertaining to lightweight concrete production; some additional properties like low-thermal conductivity and low density could nurture low-cost, medium-loaded construction activities.

As per the above-cited references, in studies, the geopolymer synthesis, comprised of RM, GBFS and FA with biomass waste, efficiently showed cost-effectiveness for geopolymer material synthesis. As given in Table 1, the minerology of the discussed industrial wastes shows promising pozzolanic efficiency on other additives, but the synergistic use of biomass waste, along with metal industrial waste, can be possible. Hence, it is possible to develop lower-cost higher-strength geopolymer concrete, in comparison to normal cement concrete.

## 8. Sustainability Indicators of Geopolymer

Numerous studies in the literature demonstrate their work on sustainable indicators, which shows the environmental and economic benefits of geopolymers. Based on the indicative performance, the sustainable indicators are broadly classified into three categories: i.e., environmental (e.g., mitigating climate change), social and economic (e.g., Operational cost, long-term saving) as discussed by Danso [63]. Environmental indicators

could include acid resistance, less embodied energy-intensive processes, groundwater rechargeable concrete as a sustainability indicator [63]. Meanwhile, for measuring sustainable geopolymer concrete/cement, the use of self-compacting concrete or the use of self-healing concrete can be used as an operational cost indicator. Moreover, the use of light-weight concrete/fire-resistant concrete water phobic/geopolymer concrete can be used as long-term savings indicator as less material consumption.

Geopolymer synthesis shows environmental and economic sustainability, based on its raw material inputs, low carbon process and the vide variety of eco-friendly material properties, as discussed above. Mehta and Kumar [64] conducted research on acid resistance, which is an environment sustainability indicator, as suggested by Danso [63]. The study evaluated the feasibility of fly ash slag-based geopolymer concrete. In this comparative study, fly ash slag-based geopolymer concrete was evaluated with normal ordinary Portland cement-based concrete. Acid suspension media of 2, 4 and 6% of sulphuric were used to perform acid resistance tests, through which the geopolymer-based concrete showed good resistance towards the acidic medium and was observed to show a 1–2% compressive strength decrement. Tempest et al. [13] carried out an embodied energy analysis of fly ash based geopolymers, which could be defined as an energy-consumption sustainability indicator. The geopolymer fly ash-based material energy requirements were compared with normal cement concrete. The geopolymer components and cement synthesis-based components were arithmetically analysed to obtain the embodied energy requirement. It was noted that the fly ash based geopolymer concrete showed 30% less embodied energy for 1 m$^3$ of production. Here, the authors stressed that the geopolymer concrete showed a smaller carbon footprint and a satisfactory relationship between the environment and economy could be achieved. An advance in geopolymer technology, with promising applications, is encouraging for the development of a special type of concrete. Chen et al. [65] investigated the development of red mud-GBFS based pervious concrete for the application of groundwater recharge, which could be classed as an environmental indicator for climate change, during the rainy season, with added quality of rainwater purification by adsorption of heavy metal ions. The observed compressive strength was up to 18.53 MPa, which seems sufficient for lightweight road pavements. The authors stressed the use of red mud for the synthesis of geopolymer concrete, which seems feasible for the function of rainwater filtered purification. Biomass and aluminium industrial waste not only contribute to strength but also to the light weight of geopolymer concrete, due to their low-density volumes. Thang Nguyen et al. [66] investigated the synthesis of lightweight geopolymer concrete, which can be classified as a long-term saving economic indicator, as expressed by Danso [63]. The geopolymer synthesis was done with the help of red mud and rice husk ash. The geopolymer concrete was developed for densities of 1205 to 1621 kg/m$^3$. The successful examination of temperature resistivity up to 1000 °C for two hours was evaluated. It was noted that the red mud-rice husk ash based geopolymer concrete showed increased strength from 36 to 166% after exposure to high temperature. This study confirms the high-temperature resistivity that can come under the category of sustainable economic indicator for operational cost of geopolymer synthesis.

Energy-saving methodologies have always been a part of lean construction practice, comprising pre- and post-construction. Patel and Shah [67] investigated the development of self-compacting geopolymer concrete, which could come under the economic indicator of operational cost to reduce extra efforts of engaged machineries. The geopolymer synthesis was done by using slag (GBFS) and rice husk ash as a raw material. The development of the self-flowing and compacting properties of geopolymer concrete has been greatly increased, due to the stabilizing properties of slag and RHA. The complete reactivity in the geopolymer medium tends to create gel-pore interactions faster, and concrete shows self-compaction. The main advantage of self-compacting concrete is faster construction and the fact that it requires less manpower, reducing the overall cost of production. This research work suggests the effective use of biomass waste, i.e., rice husk ash (RHA), for lightweight geopolymer concrete, which substantially reduces the structural load on the foundation,

enabling savings of excess materials. In terms of sustainability, the waterproof property of geopolymer concrete could come under the economic indicator of maintenance cost reduction for water proofing for the long service life of the structure. Liang et al. [68] investigated the development of waterproof geopolymer concrete with metakaolin and rice husk ash. The findings show that the use of RHA with metakaolin-based geopolymer not only improves the silica ($SiO_2$) content but also imparts strength properties in the development. The enhancement of the waterproofing property is observed, due to the formation of alumino-silicate and calcium-silicate hydrates. This microstructural property was developed because of dissolving reaction carried by alkali towards excess of silica ($SiO_2$). That enables the development of a compact microstructure and reduces inner gel spaces. This mechanism preserves the strength and enhances the softening coefficient. McGrath et al. [69] reported on Life Cycle Analysis, i.e., energy involved in the synthesis of geopolymer concrete, as an environmental indicator of fossil fuel consumption and an environmental indicator for solid waste utilization in the Malaysian context. The authors also suggest that the construction industry can possibly have a way to consume a huge amount of industrial waste, since the material consumption in construction sectors is high, involving around 8% of global $CO_2$ emissions. The comparative study of geopolymer concrete and normal cement concrete is found to be more environmentally feasible in the context of embodied energy analysis. Only the cost of activators and transportation could be involved for geopolymer concrete synthesis, unlike the case for normal concrete. According to the literature, geopolymer synthesis confirms important sustainability indicators to leverage the scope for the development of a geopolymer construction practice. Table 4 shows the sustainability indicator-based geopolymer findings for different geopolymer materials derived from aluminium, steel, biomass, and power plant industrial wastes. However, the biomass industrial waste, i.e., bagasse ash, did not perform as a special sustainable indicative geopolymer concrete.

**Table 4.** Sustainability indicators achieved by different geopolymer concretes.

| SI | Reference | Dimension | Indicator | Test/Analysis | Type of Geopolymer |
|----|-----------|-----------|-----------|---------------|--------------------|
| 1 | Mehta and Kumar [64] | Environmental [63] | Acidification effect | Acid resistance test | Fly ash and slag (GBFS) |
| 2 | Tempest et al. [13] | Environmental [63] | Fossil fuel depletion | Embodied energy analysis | Fly ash |
| 3 | Chen et al. [65] | Environmental [63] | Climate change | Groundwater recharge with water purification | Red mud |
| 4 | Thang Nguyen et al. [66] | Economic [63] | Long-term saving | Heat-resistant & lightweight | Red mud and RHA |
| 5 | Patel and Shah [67] | Economic [63] | Operational cost | Self-compacting | GBFS and RHA |
| 6 | Liang et al. [68] | Economic [63] | Maintenance cost | Waterproof | RHA and metakaolin |
| 7 | McGrath et al. [69] | Environmental [63] | Fossil fuel depletion & solid waste | Life cycle assessment | Geopolymer concrete |

## 9. Conclusions

Based on the literature review of industrial wastes or by-products, the geopolymer process can have considerable potential to utilize industrial wastes and fulfil the sustainable construction material demands. The chemical characterizations show the formation of mineralized alumino-silicate from industrial wastes like red mud, fly ash, GBFS, RHA and bagasse ash, as shown in the surface morphological characterization section. Surface and subsurface morphological characterizations demonstrate that the reactivity of supplementary cementitious materials can be improved with the use of micro-scale materials. Industrial waste based nano materials can be used as a precursor for chemical pozzolanic

to enhance the mechanical properties. Based on the studied literature, the compressive strength of geopolymer concrete has reached using fly ash 15–50 MPa, GBFS 25–70 MPa, red mud 8.8–56 MPa, RHA 20–65 and bagasse ash 11–38 MPa. To attain high compressive strength, it is vital to study the effect of influencing factors (i.e., binder ratio, molar concentration, curing temperature, slump, flexural and split tensile strength, modulus of elasticity and chemical pozzolanic activity) on the compressive strength. The durability studies for geopolymer concrete for alkali-silica reaction, acid attack and sulphate attack showed promising results up to 84 days compared to OPC based concrete. However, there are few or no long-term durability studies given in cited literature. Since alkali activators play an important role in geopolymer synthesis, the work on the molar concentration of alkali activators can aggressively proceed, in order to create cost-effective alkali activators. This technique has a flexible room to proceed for dry as well wet synthesis. The global warming potential of the cement industries always alerts the environmentalist, due to the excessive contribution to $CO_2$ emission during the process of cement production; hence, the supplementary industrial rejects can be utilized as a feed material, in order to decrease the energy requirement. The cost of alkali activators affects the energy consumption of geopolymer concrete; hence, it is necessary to find a possible low-cost alternative source. Overall, the geopolymer technique is helping in a scientific way to develop low-cost, sustainable, and desired mechanical strength construction materials to drop-down the environmental burden.

## 10. Future Research

The earlier research on scientific information about the evaluation of the chemical and physical properties of geopolymer concrete/materials is appreciated. According to the literature, the industrial waste-based geopolymer concrete approach successfully proved the evidence of geopolymer concrete having strength of up to 70 MPa. However, the metal and power industrial wastes, like red mud, GBFS and fly ash, could have further future development for the synthesis of impact-resistant ultrahigh-strength (>100 MPa) geopolymers for bunkers and military structures. These futuristic applications for special types of concrete, like blast- and abrasive-resistant geopolymers for surface/subsurface critically loaded structures, can be developed. Moreover, the use of biomass waste can work as an alkali-activated additive to pozzolanic reaction efficiency of geopolymer processes. The red mud-slag-fly ash based geopolymers show promising properties that can be further developed for high value end applications. In addition, the following areas need to be investigated in the future:

- The resource augmentation policies for geopolymer supporting industrial waste could be framed for countrywide applications. A complete standard code could be developed for geopolymer, according to the resource material origin and the mineralogical behaviour of the industrial waste.
- Since less work has been instigated for the development of fire- and blast-resistant geopolymers, to develop high-performance concrete with strength greater than 70 MPa, the work could be initiated by the implementation of nano converted industrial waste, to improve the pozzolanic reaction efficiency.
- The cost economics of the geopolymer can be calculated from various liquid industrial waste like Bayer liquor, which efficiently contains alumino-silicates.
- The development of geopolymer porous concrete could be introduced in low water table areas, in order to maintain the water level, due to geopolymer infiltration capacities.
- The global warming potential of the cement industries always alerts the environmentalist, due to the excessive contribution to $CO_2$ emission during the process of cement production; hence, the supplementary industrial rejects can be utilized as a feed material, in order to decrease the energy requirement.
- Very few works have been initiated for making geopolymers from aluminium-based industrial rejects, and many of the industrial wastes are not identified across the

different industrial sectors. Hence, the resources' efficiency and their implications could be the frame for introducing country-level geopolymer applications.

- Industrially oriented geopolymer process parameters could be set to develop area- and resource-specific products, under the theme of the circular economy concept.
- Environmentally sustainable development could be initiated by performing life cycle assessment of geopolymer products. Additionally, geopolymer, as a new construction material, can be initiated to study energy simulations and embodied energy calculation for new industrial waste-based construction materials.
- Structural dynamic studies could be instigated, to establish the technical sustainability of geopolymer materials in building and infrastructure development. Additionally, the bond behaviour of geopolymer with reinforcement could be studied, to design safe, eco-friendly, and economical construction practices.

**Author Contributions:** Both authors contributed to the preparation of the paper. N.M.A. mainly contributed to all the sections, conclusions and the answering of reviewers' questions. S.S.M.S. contributed to the abstract, analysis of cited literature, defining research questions, revising it critically for important intellectual content and conclusions. Both authors have read and agreed to the published version of the manuscript.

**Funding:** This research received no external funding.

**Institutional Review Board Statement:** Not applicable.

**Informed Consent Statement:** Not applicable.

**Data Availability Statement:** No datasets were generated or analyzed during the current study.

**Conflicts of Interest:** The authors declare no conflict of interest.

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
