# Peer review of "Utilization of Industrial By-Products/Waste to Manufacture Geopolymer Cement/Concrete"

_sustainability, doi:10.3390/su13020873_

Round 1

Reviewer 1 Report

Reviewers' comments:

Manuscript ID: Sustainability-1037666

Full Title: Utilization of industrial by-products/waste to manufacture geopolymer cement/concrete.

Comments: 

The manuscript describes the, Utilization of industrial by-products/waste to manufacture geopolymer cement/concrete. The subject of the paper is correct and interesting. But, several important points are missing and there are some points that should be revised or corrected. Some important points are mentioned hereafter.

- Abstract: the authors need to improve with more specific short results.

- Keywords: add more keywords [3 keywords is not enough for review paper].

- Introduction: should be improved; more related papers must be discussed and superiority, novelty, critical improvement in this study must be clarified. Author should add more references, 10 references is not enough for review paper.

- Figure 1. Geopolymer process diagram. Not clear make clear.  

- Line number 77: alumina (Al2O3) to alumina (Al2O3)

- Surface morphological characterization – should be detail.

- Figure 1. Flowchart depicting the general process to study disease suppressive soils (modified by the 257 author from [22]) - Make clearer.

- Conclusion - should be concise. It is too long.

- Line number 624: 1205 to 1621 kg/m3… to 1205 to 1621 kg/m3

- References: Make all references in same format for volume number, page numbers and journal name, because it is difficult to searching and reading.

- Several faults: are added or missing spaces between words: see manuscript file (For example: Line number 285: 20hrs to 20 hrs).

- Some sentences need reconstruction and the level of English should be improved.

Based on these, I advise the authors to rectify the above mentioned errors and we hope to re-evaluate the revised manuscript.

Author Response

Remark:  Modifications are being highlighted in red coloured content in manuscript.

Dear Reviewer;

Thank you for your valuable inputs. Please find the recommended corrections made for your further review.

  1. The manuscript describes the, Utilization of industrial by-products/waste to manufacture geopolymer cement/concrete. The subject of the paper is correct and interesting. But, several important points are missing and there are some points that should be revised or corrected. Some important points are mentioned hereafter.
  2. Abstract: the authors need to improve with more specific short results – Abstract have now shown with some more specific results with geopolymer properties and possible developments.
  3. Keywords: add more keywords [3 keywords is not enough for review paper] – Keywords added
  4. Introduction: should be improved; more related papers must be discussed and superiority, novelty, critical improvement in this study must be clarified. Author should add more references; 10 references is not enough for review paper – some papers have been added, with specifying the novelty of the research shown
  5. Figure 1. Geopolymer process diagram. Not clear make clear – Idea elaborated in the figure title.
  6. Line number 77: alumina (Al2O3) to alumina (Al2O3) - Corrections done – please see line number 106
  7. Surface morphological characterization – should be detail. – Corrections done
  8. Figure 1. Flowchart depicting the general process to study disease suppressive soils (modified by the 257 author from [22]) - Make clearer.- Doesn’t belongs to our manuscript I think.
  9. Conclusion - should be concise. It is too long.- Corrections rectified.
  10. Line number 624: 1205 to 1621 kg/m3… to 1205 to 1621 kg/m3 - Corrections rectified please see line number 654
  11. References: Make all references in same format for volume number, page numbers and journal name, because it is difficult to searching and reading – Corrections rectified.
  12. Several faults: are added or missing spaces between words: see manuscript file (For example: Line number 285: 20hrs to 20 hrs) - Corrections rectified.
  13. Some sentences need reconstruction and the level of English should be improved – Had worked on several sentence and reconstructed.

Please find the corrected manuscript for your further review.

Reviewer 2 Report

In this article, the authors have reviewed the utilization of industrial by-products/waste to manufacture geopolymer cement/concrete. The article is well-written. However, the novelty of this article is unclear. There are a lot of review articles on geopolymer cement/concrete. Therefore, the authors should clarify the novelty of this article in the introduction.

Author Response

Remark:  Modifications are being highlighted in red coloured content in manuscript.

Dear Reviewer;

Thank you for your valuable inputs. Please find the modified manuscript for your reference.

In this article, the authors have reviewed the utilization of industrial by-products/waste to manufacture geopolymer cement/concrete. The article is well-written. However, the novelty of this article is unclear. There are a lot of review articles on geopolymer cement/concrete. Therefore, the authors should clarify the novelty of this article in the introduction

Response : Introduction modified with possible inputs and the novelty of the work has been added.

Reviewer 3 Report

Review was written carefully and in detail. Presented tables and charts are comprehensible, and their content and data are well-organized. However, reading of text itself wasn’t quite comfortable due to its lengthiness. In my opinion, text could be briefer, and divided into more paragraphs.  Conclusion also could be more concise.

There are typo in the numbering of your references. Third table’s width exceeds the prescribed width of the pages.

Author Response

Remark:  Modifications are being highlighted in red coloured content in manuscript.

Dear Reviewer;

Thank you for your valuable inputs. Please find the modified manuscript for your further review.

1. Review was written carefully and in detail. Presented tables and charts are comprehensible, and their content and data are well-organized. However, reading the text itself wasn’t quite comfortable due to its lengthiness. In my opinion, text could be briefer, and divided into more paragraphs.  Conclusion also could be more concise. 

Response : Conclusion corrected in possible concise form.

2. There are typo in the numbering of your references. Third table’s width exceeds the prescribed width of the pages. 

Response : Reference arranged.

Round 2

Reviewer 1 Report

Reviewers' comments:

The authors revised the manuscript according to the reviewers' comments.